Accepted at the ICLR 2024 Workshop on AI4Differential Equations In Science

# Semiparametric Inference and Equation Discovery with the Bayesian Machine Scientist

**Kai-Hendrik Cohrs[1], Gherardo Varando[1], Marta Sales-Pardo[2],**
**Roger Guimerà[2,3], Gustau Camps-Valls[1]**

[1] Image Processing Laboratory, Universitat de València, València, Spain

[2] Department of Chemical Engineering, Universitat Rovira i Virgili, Tarragona, Spain

[3] ICREA, Barcelona, Spain

{kai.cohrs, gherardo.varando}@uv.es,
{marta.sales, roger.guimera}@urv.cat,
gcamps@uv.es

## Abstract

Hybrid modeling, combining machine learning with physical equations, is promising in many fields of science, in particular for climate and Earth Sciences, but faces challenges like interpretability, inconsistent extrapolation, lack of speed, and robust inference. Here we show that the *Bayesian machine scientist*, a Bayesian approach to symbolic regression is an ideal choice for the challenges in the hybrid modeling task. We formulate the *hybrid Bayesian machine scientist* and showcase its potential in the example of modeling ecosystem respiration with the $Q_{10}$ model. Specifically, we show that our proposed hybrid equation discovery method (i) extracts the correct equations, (ii) extrapolates better in different scenarios than the non-hybrid and deep-learning-based baselines, and (iii) is able to infer more accurately parameters of interest, even in the presence of equifinality. We anticipate a spur of development of hybrid equation discovery algorithms in the sciences to approach fully interpretable data-driven models.

## 1 Introduction

Earth system models (ESMs) play a major role in understanding human impact on the Earth's climate (IPCC, 2021), crucial for addressing climate change. They are complex mathematical representations of the Earth's climate system involving coupled nonlinear ordinary and partial differential equations (Scholze et al., 2012). To enhance these models, the community pursues integrations with machine learning (ML), made possible by abundant Earth observation data (Grundner et al., 2022; Reichstein et al., 2019; Camps-Valls et al., 2021). However, this approach is not without its challenges, which include violating physical laws, poor generalization to out-of-domain (OOD) data, and lack of interpretability in black-box ML models (Reichstein et al., 2019; Marcus, 2018; Quiñonero-Candela et al., 2009; Sugiyama & Kawanabe, 2012; Rudin & Radin, 2019).

Hybrid modeling, integrating ML with physical equations, has emerged as a popular approach to harness these models (Karpatne et al., 2022; Koppa et al., 2022; Zhao et al., 2019; Tramontana et al., 2020). However, challenges persist in speed, interpretability, and addressing *equifinality* (Oberpriller et al., 2021; Reichstein et al., 2022), that is the existence of multiple (parametric and non-parametric) models and sets of parameters that describe the data similarly well.

Symbolic regression is gaining attention as a promising solution to address some of the problems of ML (Camps-Valls et al., 2023). It has been shown to be effective in supporting scientific discovery in different areas of science (Abdellaoui & Mehrkanoon, 2021; Martinez-Gil & Chaves-Gonzalez, 2020; Weng et al., 2020; Wang et al., 2019; Batra et al., 2021; Lemos et al., 2022; Reichardt et al., 2020), including the Earth sciences (Grundner et al., 2023). Unlike black-box ML models, symbolic regression yields mathematical equations that are white-box models, i.e., interpretable to humans. This enables direct insight into its extrapolation behavior, providing clarity and transparency in predicting outcomes beyond the observed data range.

Because in AI-enhanced ESMs there are several sources of uncertainty, it is crucial to sample from distributions over parameter values, ML models, and, ideally, discovered equations to capture the uncertainty of its projections. A Bayesian approach is the natural choice to obtain distributions over equations (and eventually parameters). The Bayesian machine scientist (BMS) is a Bayesian formulation of symbolic regression (Guimerà et al., 2020) and has shown impressive results for extrapolation and discovery of physical equations.

In this work, we tie the ends together towards a hybrid BMS, which enables us to extract distributions over both equations and fixed meaningful parameters in a Bayesian fashion, thus pioneering the path for combining perturbed parameter ensemble (Dagon et al., 2020) with symbolic regression in climate models. Our concrete contributions are as follows: We formulate the BMS for the hybrid modeling setting and showcase its capabilities in a simple but significant problem of ecosystem respiration modeling, where the hybrid Bayesian machine scientist (HBMS): 1) improves extrapolation under different forms of OOD data; 2) recovers the true equations; and 3) shows improved inference on parameters of interest and robustness under equifinality (the lack of identifiability). We find that these results hold for both synthetic and measured data. Our findings showcase that the HBMS could be the preferred tool for offline learning in the context of ESMs.

## 2 Hybrid Bayesian Machine Scientist

Prior works have explored integrating symbolic regression and domain knowledge. Famous approaches, such as SINDy (Brunton et al., 2016) or genetic programing (GP) (Koza, 1994), have been subject to the introduction of various scientific constraints. Asadzadeh et al. (2021) demonstrated improved performance by partially fixing the regression tree in hybrid symbolic regression models with GP. Kronberger et al. (2022) introduced constraints on function image and derivatives to incorporate prior knowledge in a GP approach. Cornelio et al. (2023) developed AI-Descartes, which constraints symbolic regression using logical axioms. Additionally, Chen et al. (2021) combined sparse regression with physics-informed neural networks (PINNs) for effective use in sparse and noisy data scenarios. To our knowledge, this work is the first to apply prior knowledge constraints to the BMS, focusing on the joint inference of physical parameters in the hybrid setting with a partially fixed parametrization.

### 2.1 Inference on Parameters with the Hybrid Bayesian Machine Scientist

In the probabilistic formulation of model selection, the posterior $p(m|D)$ over models $m$ given some data $D$ encapsulates all the information about the plausibility of models. This posterior can be written as
$$p(m|D) = \frac{\exp[-\mathcal{H}(m;D)]}{Z},$$
where $\mathcal{H}(m;D)$ is the description length for jointly encoding the model $m$ and the data $D$, and $Z$ is a normalizing constant. It can be approximated as $\mathcal{H} = \text{BIC}(m;D)/2 - \log p(m)$, where BIC is the Bayesian information criterion and $p(m)$ is a suitable prior for models (Guimerà, 2020). The BMS is a symbolic regression algorithm that samples closed-form mathematical models from the posterior $p(m|D)$ using MCMC (Guimerà, 2020).

Here, we formulate the BMS for additional inference on meaningful parameters in the hybrid modeling setting, in which part of the equation is known (with some parameters $\theta$ whose values are unknown and need to be learned) and the other part $m$ needs to be inferred through symbolic regression among all possible models $M$. We obtain the posterior over $\theta$ via marginalization
$$p(\theta|D) = \sum_{m \in M} p(\theta, m|D) = \sum_{m \in M} p(\theta|m, D) p(m|D). \tag{1}$$
Since, in general, there is no closed-form solution to the posterior $p(\theta|m, D)$, we would ideally need to run a nested MCMC, sampling each equation $m$ and then, in turn, sampling from the posterior $p(\theta|m, D)$ over $\theta$ given the equation. This is computationally intensive, so we instead assume that $p(\theta|m, D)$ is peaked around some value $\theta^*$ and use the approximation
$$p(\theta|D) \approx \sum_{m \in M} \delta(\theta_m^* - \theta) p(m|D).$$
We estimate $\theta_m^*$ via maximum likelihood, which amounts to assuming a flat (enough) prior over $\theta$.

Table 1: Results on extrapolation task to OOD data scenarios. EM denotes the ensemble mean of the NN training, PPM is the posterior predictive mean, and MDL represents the model with minimum description length. RMSE and bias are given in $\mu\,\mathrm{mol}\,\mathrm{CO}_2 m^{-2} s^{-1}$.

| Model | Estimator | One year to next year | | | Night to day | | |
|-------|-----------|------|------|------|------|------|------|
| | | RMSE | R2 | Bias | RMSE | R2 | Bias |
| NN | EM | 0.049 | 0.987 | -0.073 | 0.457 | 0.944 | -0.063 |
| HNN | EM | 0.122 | 0.967 | -0.122 | 0.121 | 0.996 | -0.013 |
| BMS | PPM | 0.018 | 0.995 | 0.038 | 0.123 | 0.996 | 0.066 |
| BMS | MDL | 0.028 | 0.993 | 0.018 | 0.290 | 0.977 | 0.154 |
| HBMS | PPM | 0.013 | 0.997 | -0.008 | 0.100 | 0.997 | 0.054 |
| **HBMS** | **MDL** | **0.003** | **0.999** | **0.006** | **0.053** | **0.999** | **0.022** |

## 2.2 EXPERIMENTS

We illustrate the performance of our approach in a relevant problem in Earth sciences.

**$Q_{10}$ Model of Ecosystem Respiration** The functional relationship between temperature and respiration in terrestrial ecosystems has been classically represented via the $Q_{10}$ respiration model (Arrhenius, 1889; Van't Hoff et al., 1899; Lloyd & Taylor, 1994):

$$R_{eco}(X, T_A) = R_b(X) \cdot Q_{10}^{(T_A - 15)/10}, \tag{2}$$

where $Q_{10}$ is the parameter describing temperature sensitivity, which is the part of the equation that is well understood, $X$ is a set of meteorological drivers, and $R_b$ describes the unknown base respiration, i.e., the baseline emission of $CO_2$ of an ecosystem at the reference temperature of $15°C$. For a thorough description of the real and synthetic data, see Appendix A.1. A typical hybrid modeling approach amounts to using a neural network (NN) as an estimator for $R_b$, treating $Q_{10}$ as a trainable parameter, and fitting everything end-to-end with gradient descent, as in Reichstein et al. (2022). We will refer to this approach as hybrid Neural Network (HNN) approach in contrast to the purely non-parametric NN approach, and it will serve as a hybrid modeling baseline.

**Training-Test scenarios** We consider three scenarios: (i) 1000 randomly sampled points from the synthetic data in 2004. We test the extrapolation capabilities to the dynamics in 2005. (ii) Synthetic training points (1010 data points) according to the measurement mask of the real data for 2004. Hence, only nighttime data is used for training. At test time, we evaluate the extrapolation capabilities to the synthetic daytime data of 2004. (iii) 1010 data points measured in 2004. We test extrapolation capabilities to the measured points of 2005.

**Methodology** In all experiments, we run the BMS and the HBMS with a burn-in phase of 1000 steps and then sample 100 equations with 50 simulation steps between each sample. We report both the posterior predictive mean (PPM) and the model of minimum description length (MDL) of the runs. Similar to Reichstein et al. (2022), we deployed neural networks with two layers and a width of 16 nodes with ReLU nonlinearities in the intermediate layers and a final softmax nonlinearity. We train 100 models with Adam (Kingma & Ba, 2017) with an exponentially decaying learning rate for optimization and select the best model during the training based on $20\%$ hold-out validation data. We report the ensemble mean (EM) for these models. Finally, we run both the HBMS and HNN with $T_A$ as an additional driver to create issues of equifinality. We report extrapolation performance and inference on $Q_{10}$ for the hybrid models.

## 3 RESULTS

**HBMS extrapolates better to unseen scenarios** Initially, we assess HBMS extrapolation capabilities, focusing on two scenarios detailed in Table 1. The first involves extrapolation from one year to the next, where BMS-based approaches outperform NN-based methods. Hybrid knowledge

Table 2: Equations discovered by standard BMS and HBMS over different setups.

| Setting | Model | Discovered equation of minimum description length |
|---|---|---|
| Random synthetic | BMS | $\alpha_4(WD_{cum} - \alpha_6\alpha_3^{T_A})(\frac{SW_{POT}^{SM}}{\alpha_7} + SW_{POT}^{SM,diff}) - \alpha_7$ |
| | HBMS | $(\alpha_3 SW_{POT}^{SM,diff} + \alpha_7 SW_{POT}^{SM} + \alpha_5)\alpha_5^{WD_{cum}}Q_{10}^{0.1(T_A-15)}$ |
| Night time synthetic | BMS | $((WD_{cum} + \alpha_6 + \alpha_2 T_A)(\alpha_4 + SW_{POT}^{SM}) + SW_{POT}^{SM,diff})\alpha_0$ |
| | HBMS | $(\alpha_7 + \alpha_7 SW_{POT}^{SM,diff} + \alpha_4 SW_{POT}^{SM} + \alpha_2)\alpha_2^{WD_{cum}}Q_{10}^{0.1(T_A-15)}$ |
| Night time measured | BMS | $(\alpha_5(T_A + \alpha_4(SW_{POT}^{SM,diff})^3) + \alpha_7)SW_{POT}^{SM}$ |
| | HBMS | $\frac{-((WD_{cum} + \alpha_0^{SW_{POT}^{SM,diff}}) + SW_{POT}^{SM})}{\alpha_4}Q_{10}^{0.1(T_A-15)}$ |

enhances BMS, but harms the NN approach. In the second scenario, a shift from training on night-time data to testing on daytime data challenges pure NN extrapolation, favoring the hybrid variant. HBMS typically shows superior MDL performance, as the optimal representation beats an ensemble. On real data (refer to Table 3), PPM excels, particularly when the real mode is harder to identify. For visualizations and a brief discussion on the sampled functions, refer to Appendix A.3.

**HBMS retrieves correct and interpretable equations**   As shown in Table 2, synthetic data, the HBMS methods all picked up correct relationships of $SW_{POT}^{SM}$, $SW_{POT}^{SM,diff}$, including the exponential relationship with $WD_{cum}$. The BMS instead introduced $WD_{cum}$ linearly and found only on the random data exponential relationship. Potentially, due to the smaller range of values, a linear relationship with temperature was discovered on nighttime data alone. The HBMS found a linear relationship of $WD_{cum}$ on measured data.

**HBMS allows robust inference under equifinality**   Examining $Q_{10}$ values from the scenario with 1000 randomly sampled data points, HBMS provides a distribution closer to the actual $Q_{10}$ value of $1.5$ with the ability to capture multiple modes. HBMS remains robust even when introducing an additional temperature predictor, creating an equifinality scenario. In contrast, HNN struggles with this introduction, significantly affecting $Q_{10}$ estimation. The resilience of HBMS to the additional predictor aligns with its nature, as the introduction of unnecessary dynamics is penalized by an increase in description length.

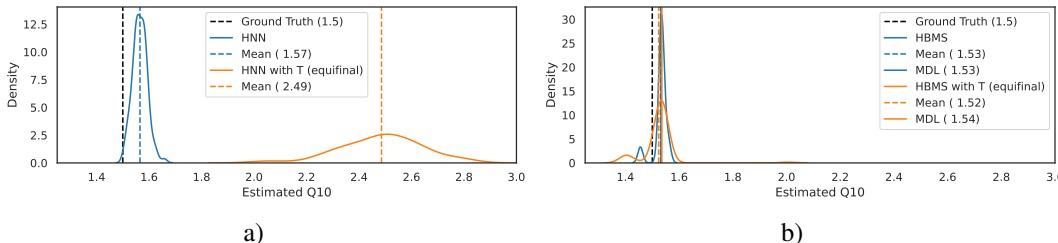

a)                                                                 b)

Figure 1: Estimated $Q_{10}$ values over 100 models of a) HNN and b) HBMS each. Results are shown for both methods without and with temperature $T_A$ as a predictor in the base respiration part $R_b$.

## 4   CONCLUSION

For advancing AI-enhanced ESMs, fast, scientifically constrained, interpretable, and probabilistic machine learning models are imperative. The proposed HBMS is a significant step forward, offering a joint distribution over parameters and interpretable equations. Our results demonstrate its superiority over non-hybrid and standard hybrid neural network baselines in extrapolation and robustness under equifinality, thus emphasizing its potential. In future work, we aim to identify more

use cases and enhance strategies for approximating the true posterior over structural parameters, potentially using methods like the Laplace approximation. Exploring additional avenues to introduce prior scientific constraints in equation discovery, facilitated by the Bayesian framework, is a crucial aspect to investigate further.

## 5 ACKNOWLEDGEMENTS

This work received support from the European Research Council (ERC) under the ERC Synergy Grant USMILE (grant agreement 855187). MS-P and RG acknowledge support from MCIN/AEI/ 10.13039/501100011033/FEDER, UE (Project No. PID2022-142600NB-I00) and by the Government of Catalonia (Project No. 2021SGR-633).

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

## A APPENDIX

### A.1 ECOSYSTEM RESPIRATION DATA

The problem of modeling ecosystem respiration is seemingly simple but essential due to the nature of $CO_2$ as a greenhouse gas, and it carries interesting caveats that make it suitable for demonstrating the capabilities of our approach. It is known that temperature is the main driver of respiration and usually has an exponential effect. How other drivers, like water availability, modulate the base respiration is highly uncertain. Another difficulty is that we can only measure it at an ecosystem level during the night; during the day, photosynthesis also affects the change in $CO_2$. The conditions for measurements of the $CO_2$ flux during the night are particularly unfavorable and high-quality measurements, hence, sparse.

**Synthetic Data** We compute the synthetic data similar to Reichstein et al. (2022) where we also take the HNN baseline from and add an additional exponential dependency to water deficit $WD_{cum}$. We use measured air temperature $T_A$, water deficit $WD_{cum}$ computed from precipitation and evaporation, and potential incoming radiation $SW_{POT}$. We compute

$$Q_{10} = 1.5, \tag{3}$$
$$R_b^{cycle} = 0.01 \cdot SW_{\text{POT}}^{\text{SM}} - 0.005 \cdot SW_{\text{POT}}^{\text{SM,diff}} \tag{4}$$
$$R_b = 0.75 \cdot (R_b^{cycle} - \min(R_b^{cycle} + 0.1\pi) \cdot \exp(0.01 WD_{cum}), \tag{5}$$
$$R_{\text{eco}} = R_b \cdot Q_{10}^{0.1 \cdot (T_A - 15)} \cdot (1 + \epsilon), \tag{6}$$
$$\tag{7}$$

where $R_b^{syn}$ describes the base respiration, which we compute with a smooth daily radiation cycle. The smooth incoming potential radiation $SW_{\text{POT}}^{\text{SM}}$ and its smoothed difference quotient $SW_{\text{POT}}^{\text{SM,diff}}$ are computed by averaging moving windows of 10 days over the incoming potential radiation $SW_{\text{POT}}$. We apply the computations in equation 5 to ensure that $R_b^{\text{syn}}$ is always positive. We sample $\epsilon$ from a centered truncated normal distribution with $0.2$ standard deviation in the interval $[-0.95, 0.95]$ to obtain heteroskedastic noise over the observations.

### A.2 REAL DATA

The performance on real data is generally not very good (see Table 3), but we extrapolate substantially better than the NN approaches. The problem with this task is that we cannot capture any anomalies outside of temperature and water availability. If someone cuts the grass from 2004 to 2005, there is no natural way for the model to pick up this change.

Table 3: Results on extrapolation task to the next year on real data.

| model | RMSE | R2 | Bias |
|-------|------|----|----|
| NN (EM) | 17.568 | 0.256 | -1.075 |
| HNN (EM) | 17.305 | 0.267 | -0.983 |
| BMS (MDL) | 15.081 | 0.361 | **-0.194** |
| BMS (PPM) | 14.732 | 0.376 | -0.276 |
| HBMS (MDL) | 14.752 | 0.375 | -0.269 |
| **HBMS (PPM)** | **14.447** | **0.388** | -0.265 |

### A.3 VISUAL INSPECTION OF SAMPLED CURVES

In the visual inspection in Figure 2, we can see that the MDL model of HBMS is almost identical to the ground truth while the HNN shows some bias from day 154 on. The HNN ensemble shows relatively uniform patterns in the uncertainty, but with the bias, the ground truth moves to the edge of the confidence bands. At the same time, the distribution of the posterior predictive of the HBMS

visually seems to capture the uncertainty better when the predictive mean deviates from the true curve.

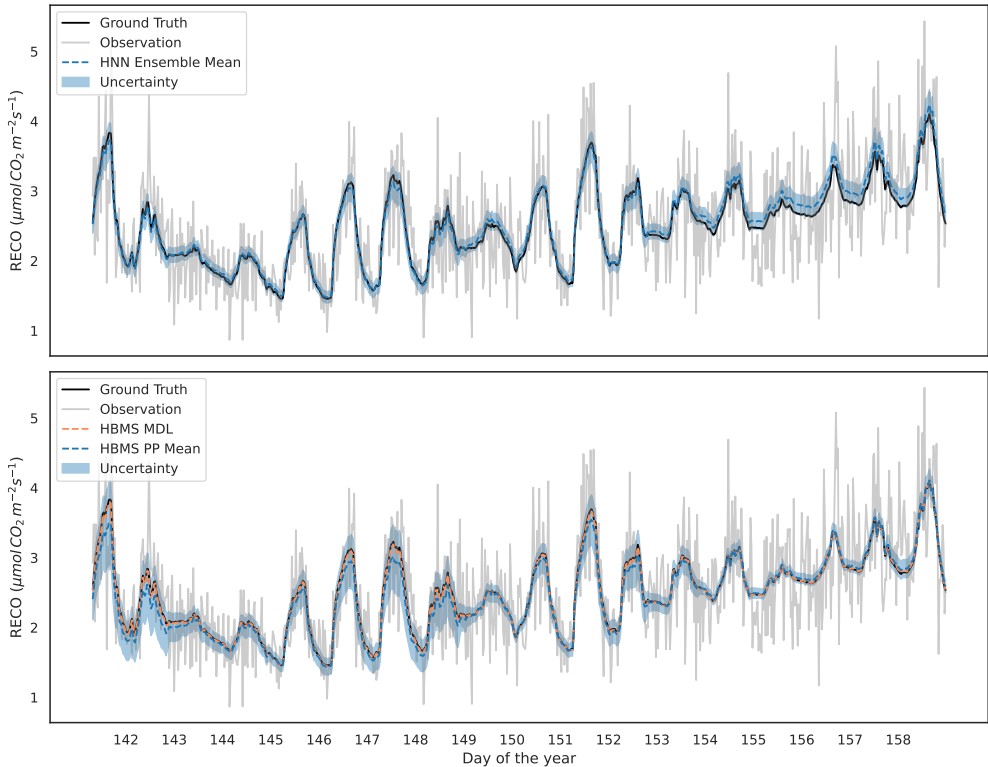

Figure 2: Ensemble plots of both hybrid modeling approaches in the extrapolation task from night-time to daytime.

