# OpenReview forum: "Semiparametric Inference and Equation Discovery with the Bayesian Machine Scientist"
_ICLR.cc/2024/Workshop/AI4DiffEqtnsInSci — AI4DiffEqtnsInSci @ ICLR 2024 Poster_

### Official Review · Reviewer_sJBC · 2024-02-14
**Innovative method for symbolic equation discovery**

**Rating:** 6
**Confidence:** 4

**Review:**

The paper showcases an innovative approach for symbolic equation discovery, a topic that has gained a lot of relevance. The authors show improved results in benchmarking of their method compared to the original BMS method but lack comparisons to other well established methods for symbolic discovery. The paper is well written and has a focus in earth sciences with results shown for the models of the domain.

---

### Meta-Review · Area_Chair_j7BG · 2024-02-29

**Recommendation:** Accept (Poster)

**Metareview:**

This paper shows that the Bayesian machine scientist, a Bayesian approach to symbolic regression for hybrid modeling tasks. It is expected that authors will be addressing comments by the reviewer in the camera ready version

---

### Decision · Program_Chairs · 2024-02-29

Accept (Poster)